# Lipoprotein(a) as a Risk Factor for Cardiovascular Diseases: Pathophysiology and Treatment Perspectives

**DOI:** 10.3390/ijerph20186721

**Published:** 2023-09-06

**Authors:** Pierandrea Vinci, Filippo Giorgio Di Girolamo, Emiliano Panizon, Letizia Maria Tosoni, Carla Cerrato, Federica Pellicori, Nicola Altamura, Alessia Pirulli, Michele Zaccari, Chiara Biasinutto, Chiara Roni, Nicola Fiotti, Paolo Schincariol, Alessandro Mangogna, Gianni Biolo

**Affiliations:** 1Clinica Medica, Cattinara Hospital, Department of Medical Surgical and Health Science, University of Trieste, 34149 Trieste, Italy; fgdigirolamo@units.it (F.G.D.G.); emiliano.panizon@asugi.sanita.fvg.it (E.P.); letizia.tosoni@gmail.com (L.M.T.); arancioca@hotmail.it (C.C.); federica.pellicori@live.it (F.P.); altamuranic@yahoo.it (N.A.); alessia.pirulli@asugi.sanita.fvg.it (A.P.); mzaccari00@gmail.com (M.Z.); fiotti@units.it (N.F.); biolo@units.it (G.B.); 2SC Assistenza Farmaceutica, Cattinara Hospital, Azienda Sanitaria Universitaria Integrata di Trieste, 34149 Trieste, Italy; chiara.biasinutto@gmail.com (C.B.); chiara.roni@asugi.sanita.fvg.it (C.R.); paolo.schincariol@asugi.sanita.fvg.it (P.S.); 3Institute for Maternal and Child Health, I.R.C.C.S “Burlo Garofolo”, 34137 Trieste, Italy; alessandro.mangogna@burlo.trieste.it

**Keywords:** lipoprotein(a), pharmacological approach, cardiovascular risk, metabolism, body composition

## Abstract

Cardiovascular disease (CVD) is still a leading cause of morbidity and mortality, despite all the progress achieved as regards to both prevention and treatment. Having high levels of lipoprotein(a) [Lp(a)] is a risk factor for cardiovascular disease that operates independently. It can increase the risk of developing cardiovascular disease even when LDL cholesterol (LDL-C) levels are within the recommended range, which is referred to as residual cardiovascular risk. Lp(a) is an LDL-like particle present in human plasma, in which a large plasminogen-like glycoprotein, apolipoprotein(a) [Apo(a)], is covalently bound to Apo B100 via one disulfide bridge. Apo(a) contains one plasminogen-like kringle V structure, a variable number of plasminogen-like kringle IV structures (types 1–10), and one inactive protease region. There is a large inter-individual variation of plasma concentrations of Lp(a), mainly ascribable to genetic variants in the Lp(a) gene: in the general po-pulation, Lp(a) levels can range from <1 mg/dL to >1000 mg/dL. Concentrations also vary between different ethnicities. Lp(a) has been established as one of the risk factors that play an important role in the development of atherosclerotic plaque. Indeed, high concentrations of Lp(a) have been related to a greater risk of ischemic CVD, aortic valve stenosis, and heart failure. The threshold value has been set at 50 mg/dL, but the risk may increase already at levels above 30 mg/dL. Although there is a well-established and strong link between high Lp(a) levels and coronary as well as cerebrovascular disease, the evidence regarding incident peripheral arterial disease and carotid atherosclerosis is not as conclusive. Because lifestyle changes and standard lipid-lowering treatments, such as statins, niacin, and cholesteryl ester transfer protein inhibitors, are not highly effective in reducing Lp(a) levels, there is increased interest in developing new drugs that can address this issue. PCSK9 inhibitors seem to be capable of reducing Lp(a) levels by 25–30%. Mipomersen decreases Lp(a) levels by 25–40%, but its use is burdened with important side effects. At the current time, the most effective and tolerated treatment for patients with a high Lp(a) plasma level is apheresis, while antisense oligonucleotides, small interfering RNAs, and microRNAs, which reduce Lp(a) levels by targeting RNA molecules and regulating gene expression as well as protein production levels, are the most widely explored and promising perspectives. The aim of this review is to provide an update on the current state of the art with regard to Lp(a) pathophysiological mechanisms, focusing on the most effective strategies for lowering Lp(a), including new emerging alternative therapies. The purpose of this manuscript is to improve the management of hyperlipoproteinemia(a) in order to achieve better control of the residual cardiovascular risk, which remains unacceptably high.

## 1. Introduction

Cardiovascular disease (CVD) remains a significant global health challenge, despite considerable advancements in its prevention and treatment. One crucial risk factor associated with CVD is elevated levels of lipoprotein(a) [Lp(a)], which operates independently and contributes to what is known as residual cardiovascular risk. Remarkably, even when low-density lipoprotein cholesterol (LDL-C) levels fall within recommended ranges, high Lp(a) can still increase the risk of developing cardiovascular events. Lp(a) is a unique LDL-like particle found in human plasma, comprising a large plasminogen-like glycoprotein, called apolipoprotein(a) [Apo(a)], covalently bound to Apo B100 via a disulfide bridge. Apo(a) consists of plasminogen-like kringle V and variable kringle IV structures (up to 40 copies per allele), along with an inactive protease region. The plasma concentrations of Lp(a) exhibit significant inter-individual variability, primarily attributed to genetic variants in the Lp(a) gene, resulting in levels ranging from <1 mg/dL to >1000 mg/dL in the general population. Moreover, Lp(a) concentrations vary among different ethnicities.

Lp(a) has emerged as a crucial risk factor implicated in the development of atherosclerotic plaque. Higher Lp(a) concentrations have been associated with an increased risk of various cardiovascular conditions, including ischemic CVD, aortic valve stenosis, and heart failure. A threshold value of 50 mg/dL has been established for clinical significance, but even levels above 30 mg/dL may escalate the risk. While a robust and well-established link exists between elevated Lp(a) levels and coronary as well as cerebrovascular diseases, the evidence concerning incident peripheral arterial disease (PAD) and carotid atherosclerosis remains less conclusive. Standard lifestyle modifications and lipid-lowering therapies, such as statins, niacin, and cholesteryl ester transfer protein (CETP) inhibitors, have demonstrated limited efficacy in reducing Lp(a) levels. This limitation has spurred increased interest in developing novel drugs to address this issue.

Among potential therapeutic avenues, PCSK9 inhibitors have shown promise in reducing Lp(a) levels by 25–30%, offering a potential breakthrough. Another approach involves mipomersen, an antisense oligonucleotide (ASO) that can reduce Lp(a) levels by 25–40%, but its use is hampered by significant side effects. Currently, lipoprotein apheresis stands out as the most effective and well-tolerated treatment for individuals with high Lp(a) plasma levels. This extracorporeal method selectively removes lipoproteins containing Apo B100, resulting in an over 50% reduction in atherogenic lipoprotein. In addition to apheresis, emerging therapies involving antisense oligonucleotides (ASOs), small interfering RNAs (siRNAs), and microRNAs hold great promise in reducing Lp(a) levels by targeting RNA molecules and regulating gene expression as well as protein production levels.

The aim of this comprehensive review is to provide an updated insight into the pathophysiological mechanisms of Lp(a), with a particular focus on the most effective strategies for lowering Lp(a) levels, including the investigation of new and emerging alternative therapies. By addressing hyperlipoproteinemia(a) more effectively, this review aims to contribute to the better management of residual cardiovascular risk, which continues to present a significant challenge in clinical practice.

### Search Strategy

The authors searched three bibliographic databases—PubMed, MEDLINE, and Embase—from the inceptions of the databases until Jul 2023, using a combination of terms related to Lp(a) prevalence, risk factors for cardiovascular diseases, pathophysiology, and treatment, without restrictions on language or publication date.

## 2. Hyperlipoproteinemia(a), the Hidden CV Risk Factor

Despite all of the progress achieved in terms of prevention and treatment, CVDs remain the primary cause of morbidity and mortality in the world population. Indeed, even if all of the best evidence-based strategies are applied (such as tailored pharmacological therapy plus lifestyle modifications, leading to the full achievement of goals for LDL-C reduction and blood pressure as well as glycemia normalization, according to the current guidelines), the observed residual cardiovascular (CV) risk remains high. Therefore, independent risk factors should be considered. Hyperlipoproteinemia(a) represents a widespread health problem in the global population: indeed, levels of Lp(a) >50 mg/dL have been found in 10–30% of the world population, with an estimated 1.43 billion people affected in the world, of which 148 million are in Europe [1] (Table 1).

### 2.1. Structure and Metabolism

Lp(a) is a low-density lipoprotein (LDL) variant where a large plasminogen-like glycoprotein, i.e., apolipoprotein(a), or Apo(a), is covalently bounded to Apo B100 via one disulfide bridge. Apo(a) contains one plasminogen-like kringle V structure, a variable number of plasminogen-like kringle IV structures (types 1–10), and one inactive protease region [2] (Figure 1). Lipoprotein (a) is synthesized in the liver [3]; its catabolism, however, is not completely understood [4]. Indeed, Lp(a) has a longer plasma half-life than LDL, suggesting a distinct metabolic pathway involved in its degradation. Because of its configuration and composition, Lp(a) is able to interact with different receptors, such as the LDL receptor protein megalin, very-low-density lipoprotein (VLDL) receptor, galactose-specific asialoglycoprotein receptor (ASGPR), plasminogen receptor, and macrophage receptors [5].

### 2.2. The Role of Genetic and “Non-Genetic” Factors in Lp(a) Metabolism

Lp(a) plasma concentration shows a wide inter-individual variation in the general population, ranging from <1 mg/dL to >1000 mg/dL [6]. Nonetheless, Lp(a) plasma concentration remains relatively constant over the course of life in men, while in women it tends to increase with age after menopause, with levels peaking during late peri- and post-menopause [7]. Lpa(a) concentrations also differ among ethnicities: it tends to be higher in individuals of African descent when compared to populations with European or Asian heritage [8,9]. Some authors have hypothesized that the differences among populations may be due to different Lp(a) gene variants [9]. Therefore, a higher Lp(a) plasmatic concentration could be considered a genetic feature largely controlled by genetic variants of the Lp(a) gene. The heritability of Lp(a) has been shown to be very high by twin and family studies, assessed as being from 70% to over 90% [10,11,12]. Indeed, differently from other lipoprotein concentrations, Lp(a) is minimally responsive to lifestyle or behavior changes, suggesting that Lp(a) levels are mostly genetically determined [13].

The gene responsible for producing Lp(a) is situated at positions 26 and 27 on the long arm of chromosome 6 (6q26-27). This gene is characterized by a high degree of polymorphism and consists of a variable exonic repeat. The gene encodes a protein domain referred to as “kringle” (k) [14], which evolved from the plasminogen gene (*PLG*). The plasminogen gene contains one protease domain and five different types of kringle domains. In about 40 million years the PLG gene has been remodeled into the Lp(a) gene; in the process it lost KI, KII, and KIII, kept only one copy of KV, and its protease domain lost activity [15,16].

The most important predictors of Lp(a) levels are an array of *LPA* genetic polymorphisms that mainly account for different isoforms of the protein, which contribute to the variation in Lp(a) concentrations in a range of 40% to 70% [17]. In fact, polymorphisms of the Lp(a) gene result in lipoprotein isoforms of different sizes, and studies have revealed that the size of Apo(a) changes in relation to kringle IV type 2 (KIV-2) copy number variations, with a clear inverse correlation between the number of KIV-2 repeats and Lp(a) concentration [18,19,20]. According to the Copenhagen General Population Studies, up to 27% of the Lp(a) concentration variation depends on the population studied, but also on the Lp(a) gene KIV-2 repeats number. Beyond KIV-2 repeats, a high Lp(a) plasma level is also associated with the pentanucleotide repeat polymorphism (PNRP) of the promoter 5′ control region of the Apo(a) gene, as well as several single-nucleotide polymorphisms (SNPs) [21,22]. To date, only a few SNPs (including rs3798220 and rs10455872 SNPs) have shown a significant association with increased plasma Lp(a) levels and CVD [23]. Nonetheless, the fact that multiple SNPs with different, mostly small, effect sizes, associated with both high and low Lp(a) levels, may contribute collectively to the variance of the trait should be considered [24]. Obviously, the contribution of other genes requires further investigations.

Although plasma Lp(a) concentration levels are strongly genetically determined, some evidence suggests that nongenetic factors, including renal function, gender, hormones, and inflammation, may also affect Lp(a) [25]. The kidney has been shown to play a role in Lp(a) catabolism, and Lp(a) levels are associated with kidney disease [26]. Several studies have also evaluated how Lp(a) concentration is affected by hemodialysis in patients with end-stage renal disease (ESRD). This clinical condition results in Lp(a) levels 5 to 10 times higher than those in patients with early-stage renal disease. Furthermore, Lp(a) concentration returns to normal levels after a kidney transplantation, probably because of improved clearance [27]. With regard to the hormonal regulation of Lp(a) levels, it has been observed that they are usually higher in postmenopausal women, suggesting an estrogen-related inhibiting effect [28,29]. Lp(a) has been studied in both normal pregnancies and in those complicated by various pathological conditions, such as pre-eclampsia, showing that there is a two-fold increase in Lp(a) during normal pregnancy [30]. Androgens, such as endogenous testosterone levels, may affect Lp(a); men with low testosterone levels have higher Lp(a) levels, and they seem to have a greater risk of heart disease [31,32]. Furthermore, growth hormone (GH) has been linked to a strong Lp(a)-stimulating effect, both in patients with acromegaly and in GH therapy [33,34]. Thyroid hormones also seem to have a modulatory effect on Lp(a), but the mechanism is still unclear. Acute inflammatory states, including sepsis, inflammatory bowel disease, gallbladder fistula, and acute myocardial infarction, have been shown to increase Lp(a) concentration [35]. Furthermore, a direct correlation has been found between the plasma concentration of Lp(a) and some inflammatory proteins, such as IL-6, CRP, and α 1 antitrypsin [36]. The relationship between Lp(a) and diabetes remains unclear: prospective findings demonstrate a strong inverse association between Lp(a) levels and the risk of developing type 2 diabetes [37,38]. On the other hand, data collected by several studies show that ethanol and tobacco consumption, regardless of different Apo(a) isoforms, reduce, on a dose-dependent basis, Lp(a) concentration by up to 60% and 20%, respectively [39,40,41,42]; the mechanism behind this phenomenon is, though, still unclear.

## 3. Pathophysiology of Lp(a)

As of now, a clear physiological function of Lp(a) has not been clearly recognized, but the homology between Apo(a) and PLG suggests that Lp(a) may represent a link between cholesterol transport and the fibrinolytic system, with a role in the modulation of blood clotting and the fibrinolytic process. Indeed, Lp(a) could enhance wound healing by carrying cholesterol to an injury site and play a role in hemostasis via the inhibition of fibrinolysis [43].

When there is an injury or wound in the body, Lp(a) may play a role in the healing process by transporting cholesterol to the site of the injury. Cholesterol is essential for cell membrane formation and repair, and it also serves as a building block for various molecules involved in tissue repair and regeneration. Therefore, the presence of Lp(a) at the wound site may aid in providing the necessary cholesterol for the healing process [44].

Lp(a) has been suggested to play a role in hemostasis via inhibiting fibrinolysis. In other words, Lp(a) may help stabilize blood clots by preventing their premature breakdown, which could be beneficial in situations where maintaining a stable clot is necessary for effective wound healing and preventing excessive bleeding [45].

It is important to note that, while these potential roles of Lp(a) in wound healing and hemostasis have been proposed, research on this topic is ongoing, and the exact mechanisms and implications are not fully understood.

An elevated Lp(a) level is a strong, causal, and independent risk factor for CVD through multiple pathogenetic mechanisms: proatherogenic, prothrombotic, and proinflammatory [46].

Experimental evidence from in vitro studies and animal models shows Lp(a)’s promotion of atherosclerotic plaque formation through various mechanisms, such as smooth muscle cell proliferation, foam cells formation, and increased expression of IL-8, a key mediator of plaque formation, from inflammatory cells [47,48]. Lp(a) can interact with the major components of the extracellular matrix (fibrin, fibronectin, proteoglycans, tretranectin, and beta2-glycoprotein) [49]. The binding of Lp(a) to fibrin has been proposed as the mechanism that allows for the delivering of cholesterol to sites of injury, where it is involved in the repairing of the vascular wall. It has also been observed that Lp(a) is able to bind oxidized phospholipids (OxPls) in plasma by forming covalent bonds. Levels of Lp(a) and OxPls in plasma predicted the risk of the incidence of CVD [50].

In order to cause CVD, Lp(a) needs to be able to enter into the intima of arteries as well as aortic valve leaflets and accumulate. Studies on human and rabbit kinetics have revealed that Lp(a) can enter normal and atherosclerotic arteries at a comparable rate to LDL [51]; however, Lp(a) has a tendency to accumulate throughout the intima, whereas the accumulation of LDL cholesterol and other lipoproteins containing apoB is typically restricted to atherosclerotic lesions. It has been observed that Lp(a) accumulates two- to three-fold greater than LDL in the arterial wall at sites of injury (balloon-injured rabbit arteries). Lp(a) and Apo(a) have also been found in end-stage aortic valve stenosis and in lesioned intima of the coronary artery bypass [52,53,54]. Lp(a) appears to have a greater capacity than LDL to bind fibrin and glycosaminoglycans, two proteins exposed at sites of injury [55,56]. In conclusion, it is possible that the intimal accumulation of Lp(a) is driven by a different mechanism than that involving LDL and other apoB-containing lipoproteins.

## 4. Laboratory Assessment

Lp(a) measurements are mainly obtained by immunoassays; however, some major issues can affect the accuracy of Lp(a) quantification and consequently the clinical interpretation of the data. Size variability of Apo(a) isoforms is one of the most critical problems, since it may lead to the over- or under-estimation of Lp(a) levels [57]. In fact, there are two types of immunoassays with which to determine Lp(a) concentration: those that are “isoform-dependent” and those that are “isoform-independent”. Lp(a) concentrations have historically been described as total mass concentration (mg/dL). In the 1970s, Albers et al. purified Lp(a) from plasma; they individually measured protein, lipid, and carbohydrate components, and they used the sum of the components as an assay calibrator, with a value in mg/dL. All of the subsequent immunoassays were then calibrated in mg/dL, although they only measured the Apo(a) component and not the total Lp(a) mass [58]. Indeed, the “isoform-dependent” method uses monoclonal or polyclonal antibodies in order to evaluate the whole lipoprotein mass (mg/dL), and it is highly related to the KIV-2 copies number. At present it is well known that, with this method, Lp(a) concentrations are overestimated in patient samples containing large isoforms and underestimated in samples containing small isoforms; however, recent methods that use five independent calibrators with a large range of Lp(a) levels and distribution of apo(a) isoforms are able to reduce the confounding impact of Apo(a) size if the values of the assay calibrators are well validated [59]. Although this method shows higher precision, it does not completely eliminate the Apo(a) isoform bias. On the other hand, the isoform-independent method (nmol/L) uses monoclonal antibodies able to bind KIV-9, the unique non-variable Apo(a) domain, reflecting the number of Lp(a) particles.

The ELISA method was approved to be the gold standard with which to measure Lp(a) concentration by the World Health Organization (WHO), and it is calibrated in nmol/l, as has been standardized by the International Federation of Clinical Chemistry and Laboratory Medicine (IFCC) [60,61,62].

The two immunoassay approaches report Lp(a) levels using two different units: mg/dL and nmol/L. Indeed, Lp(a) concentrations should not be converted from nmol/L into mg/dL, or vice versa, as all conversion factors are isoform-dependent [61,63]. Previous studies have tried to convert mg/dL or mg/L into nmol/L by using a correction factor equal to 2.4, but this approach is not recommended since it does not consider the wide heterogeneity of Apo(a) type. The existence of two different units for expressing Lp(a) levels can be confusing for both clinicians and patients.

Nowadays, none of the current commercially available assays for Lp(a) measurement are completely inherently isoform-insensitive [62,64]. Innovation in the development of isoform-insensitive Lp(a) assays is currently an unmet need. The development of non-antibody-based methods may provide the right approach with which to avoid the bias from the size variability of Apo(a). There are currently three methods with which to assess the KIV repeats number and Apo(a) isoform size at the DNA level: pulsed field gel electrophoresis (PFGE) and fiber-fluorescence in situ hybridization (FISH), which allow the evaluation of KIV-2 repeats number in separate alleles, while the quantitative polymerase chain reaction (qPCR) evaluates the sum of KIV-2 copies in examined genomes [65].

## 5. Lp(a) in Vascular Diseases

### 5.1. Cardiovascular Risk

Coronary heart disease (CHD) is a major CVD, and it is one of the leading causes of death in both developed and developing countries [66,67]. CHD is an atherosclerotic disease whose clinical manifestations include stable angina, unstable angina, myocardial infarction (MI), or sudden cardiac death [68]. Modifiable known risk factors for CAD include diabetes mellitus, hypertension, smoking, hyperlipidemia, obesity, chronic kidney disease, hyperhomocysteinemia, and psychosocial stress [69].

High Lp(a) concentrations have been consistently linked to an increased risk of ischemic CVD, aortic valve stenosis, and heart failure. Both the American Heart Association/American College of Cardiology (AHA/ACC) and the European Society of Cardiology (ESC) guidelines have suggested a threshold of 50 mg/dL of Lp(a) to mitigate an increased risk of CVD; however, some experts believe that the risk of CVD may already increase when Lp(a) plasma levels exceed 30 mg/dL [70]. The 2019 ESC/EAS guidelines recommend that individuals undergo at least one Lp(a) plasma level test in their lifetime. It is worth noting that individuals with congenitally high Lp(a) levels (e.g., >180 mg/dL) may have an ASCVD (atherosclerotic CVD) risk equivalent to those with a family history of premature cardiovascular disease or those with heterozygous familial hypercholesterolemia [71]. Although there is no clear biochemical mechanism to explain how higher Lp(a) levels increase the risk of CHD, researchers have proposed several potential mechanisms, including the involvement of LDL-C [72], inhibition of the conversion of plasminogen into plasmin [73], and the ability to transport proinflammatory oxidized phospholipids as well as disseminate them as carriers [74].

To contribute to CVD, Lp(a) must have the ability to accumulate in the intima of arteries. This accumulation in locations of vascular damage is thought to be one of the primary mechanisms through which elevated Lp(a) leads to CVD [73,74]. Additionally, research suggests that Lp(a) may also contribute to foam cell development through macrophage uptake, a mechanism that is also associated with LDL and triglyceride-rich lipoproteins, as well as the development of atherosclerosis [75,76,77].

A meta-analysis conducted by Wendy Craig and colleagues in 1998 found that Lp(a) concentrations were higher in individuals who developed ischemic heart disease compared to those who did not [78]. This conclusion was based on data from population-based prospective cohort studies and nested case–control studies [79,80,81,82,83].

John Danesh and colleagues confirmed a clear association between Lp(a) and CHD by updating this meta-analysis with an added follow-up of 10 years inclusive of 5436 additional deaths from CHD [84]. The INTERHEART study and the Copenhagen City Heart Study examined the contribution of Lp(a) concentration to MI risk in the general population and assessed that concentrations of Lp(a) >50 mg/dL were associated with an increased risk of MI [85].

Even large Mendelian randomization studies, which are free from confounding bias and reverse causation, strongly support the notion that elevated Lp(a) represents an independent, genetic causal factor of CVD [86,87]. Although the genomic determinants of Lp(a) and their impact on the risk of coronary disease are not well understood, the argument for direct, genetic causality in CVD is much more compelling for Lp(a) than for the majority of other cardiovascular risk factors [88]. A case–control study recruited 3145 case subjects with coronary artery disease and 3352 control subjects from four European countries; in this study, two common Lp(a) variants (rs10455872 and rs3798220) were shown to be strongly associated with both an increased level of Lp(a) lipoprotein [explaining together 36% of the total variation in Lp(a) concentration] and an increased risk of coronary disease [89].

### 5.2. Aortic Valve Stenosis

Calcific aortic valve disease (CAVD) is a progressive condition that results from the severe calcification of the aortic valve (i.e., aortic valve stenosis, AS), which has three leaflets and regulates blood flow from the heart. This calcification hinders the valve’s movement and restricts the outflow of blood from the ventricle. CAVD affects about 2% of people over 65 years old, and symptoms usually do not appear until the disease is advanced. Epidemiological research has revealed that high levels of Lp(a) in the blood increase the risk of AS. A study from the 1990s found that genetic variations in the Lp(a) locus were linked to aortic valve calcification, and several studies have shown that an rs10455872 Lp(a) SNP doubles the risk of aortic valve calcification in various ethnic groups [90]. Furthermore, the Lp(a) genotype is associated with an increased risk of aortic valve replacement surgery and degree of aortic valve calcification [91,92,93]. These findings are reported in Figure 2.

Imaging studies (based on both computed tomography calcium scoring and echocardiography) observed increased valve calcification and faster rates of disease progression in elderly patients with high Lp(a) levels [94]. Recently, this association was not confirmed by Kaiser and colleagues, who assessed that Lp(a) might play a role in the initial phase of AS but found no association between Lp(a) levels and the progression of disease; if this evidence is confirmed, the effectiveness Lp(a) of Lp(a)-lowering in affecting the clinical outcomes of AS may be limited to pre-calcific stages of aortic valve disease [95].

The epidemiological and genetic evidence that high Lp(a) levels are linked with the development of AS is quite strong. Lp(a) is an important carrier of oxidized phospholipids (OxPL) that may play a key role in the process [96]. Moreover, Lp(a) may favor the onset and development of CAVD by causing aortic valve endothelial dysfunction, accumulating in the valve, and delivering its OxPL content along with autotaxin (ATX) [97]. This mechanism promotes not only inflammation but also the osteogenic transformation of valvular interstitial cells causing calcium deposition [98]. Valve calcification is characterized by an inflammatory response, which involves the secretion of lipoprotein-associated phospholipase A2 (Lp-PLA2) by macrophages. Lp-PLA2 utilizes OxPLs as a substrate and produces lysophosphatidylcholine (LPC), which is then converted by ATX into lysophosphatidic acid [Lp(a)]. Lp(a) binds to and activates the Lp(a) receptor [Lp(a)R], triggering the activation of NF-κB, leading to increased inflammation and secretion of interleukin-6 (IL-6). This process results in osteogenic differentiation, valve leaflet calcification, and eventually AS [99,100]. Additionally, Apo(a), B, and E have been found in aortic valve lesions [101]. Despite Lp(a) consistently being higher in black individuals than in Caucasians, the prevalence of AS is higher in Caucasians than in Blacks, Hispanics, and Asians in the US [102].

While elevated Lp(a) is acknowledged as a major predictive factor for clinical AS in patients with heterozygous familial hypercholesterolemia (FH) [96], its influence on aortic stenosis in the general population remains to be fully determined.

### 5.3. Peripheral Arterial Disease

Peripheral arterial disease (PAD) is an atherosclerotic disorder characterized by a progressive narrowing, eventually to the point of occlusion, of large- and medium-sized arteries of the extremities, especially the lower ones [103,104]. PAD is largely diffuse worldwide, and it is associated with increased CV morbidity as well as mortality. As in other atherosclerotic diseases, many risk factors have been established as playing an important role in plaque formation and progression. In addition to traditional risk factors, such as hypertension, hypercholesterolemia, and diabetes mellitus, Lp(a) has also been investigated as a risk factor [105].

Despite the strong and well-established association between elevated Lp(a) levels and coronary as well as cerebrovascular disease, data regarding incident PAD and carotid atherosclerosis are less robust: most studies show a direct correlation between Lp(a) levels and PAD, while in others this association is not definitely confirmed [106].

Genetic, pathophysiologic, and epidemiologic studies, such as the Copenhagen General Population Study, support the notion of Lp(a) as a potential causative factor in femoral stenosis [107]. In the most recent GWAS study, Klarin et al. identified Lp(a) variants strongly associated with PAD [108]. A different study, by Lashkolnig et al., showed a significant association between Lp(a) concentrations low-molecular-weight (LMW) Apo(a) phenotypes, and rs10455872 with PAD, both symptomatic and asymptomatic [109]. Both the MESA study and the InCHIANTI study demonstrated a notable correlation between elevated Lp(a) levels and PAD in the lower extremities [110,111]. The EPIC-Norfolk prospective study also showed an increased risk in terms of a 1.37 HR for PAD per a 2.7-fold increase in Lp(a) levels, an association that proved to be independent of LDL cholesterol levels [112].

Despite the ESC and ACC/AHA guidelines stating that the risk of CVD is significant when Lp(a) >50 mg/dL [113,114], in several studies the cut-off level of Lp(a) for a potentially increased risk of PAD was set to 30 mg/dL. A small study reported that Lp(a) levels greater than 30 mg/dL are associated with a 3.9-fold increase in the risk of premature PAD [115]. Another cross-sectional study involving 557 patients with type 2 diabetes showed similar results, with Lp(a) levels exceeding 30 mg/dL carrying a threefold higher risk of PAD. This study also found an inverse correlation between Lp(a) levels and the ankle–brachial index (ABI) [116]. In a prospective case–control study, Lp(a) levels above 24 mg/dL were linked to a two-fold increased risk of PAD. Furthermore, higher levels of Lp(a) were associated with more severe forms of PAD [117]. A higher risk for ischemic stroke, MI, or limb amputation has been observed in symptomatic PAD patients with Lp(a) >30 mg/dL, compared to similarly symptomatic patients with lower levels [118]. A recent study showed that patients with PAD and Lp(a) >30 mg/dL had a greater need for any PAD operation [119]. Yanaka et al. also showed that a higher Lp(a) level was an independent predictor for the loss of primary patency after endovascular therapy (EVT) [120].

A recent systematic review analyzed 15 studies involving 493,650 patients, and the majority of these studies supported a significant association between high Lp(a) levels and the risk of PAD, according to the authors. High Lp(a) levels were also found to be linked to an increased risk of claudication, PAD progression, restenosis, hospitalization, and death [121] (Figure 2).

### 5.4. Carotid Atherosclerosis

In 2020, globally, around 28% of individuals aged 30–79 years in the general population were found to have an abnormal carotid intima-media thickness of 1.0 mm and above, which corresponds to slightly over one billion people. Moreover, approximately 21% of people aged 30–79 years had carotid plaque, while 1.5% had carotid stenosis, indicating roughly 816 million individuals with carotid plaque and 58 million with carotid stenosis [122]. Lp(a) has been established as one of the risk factors that play an important role in the development of atherosclerotic plaque [123]. Increased Lp(a) levels seem to be associated with carotid artery stenosis: Klein et al. reported that Lp(a) was an independent predictor of stenosis and occlusion, but not of carotid plaque area [124]. Different studies have shown that elevated Lp(a) levels were independent predictors of increased carotid atherosclerotic burden, and other epidemiological as well as genetic studies have shown a continuous and independent association between Lp(a) and cerebrovascular disease [125]. Lp(a) seems, indeed, to be significantly correlated with stenosis and occlusion, which are frequently the consequences of plaque rupture and thrombosis. This may lead to the hypothesis that the role of Lp(a) in atherogenesis may be largely based on its effect on coagulation and thrombosis.

In patients who have suffered an ischemic stroke, elevated Lp(a) levels are associated with the presence of carotid atherosclerosis [126]. Furthermore, in the AIM-HIGH study, Lp(a) was associated with high-risk plaque features, such as the presence of a mural thrombus, intraplaque hemorrhage, or surface defects [127]. Elevated Lp(a) levels may independently predict the risk of carotid atherosclerosis progression, despite a strict LDL-C control [128].

Contrary to the results mentioned previously, Ooi and colleagues’ research revealed that while plasma Lp(a) concentration was independently linked to the extent and severity of coronary artery disease (CAD) upon angiography, it did not display a significant association with carotid artery plaque. This suggests that the impact of plasma Lp(a) levels may differ between the two vascular conditions [129].

Gender-associated differences in Lp(a) distribution, carotid plaque composition, and the frequency of vulnerable plaques have been shown [130]: while men tend to have more vulnerable plaques than women, a recent study has revealed that in women elevated plasma Lp(a) levels were associated with a higher prevalence of intraplaque hemorrhage (IPH), while in men they were associated with a higher degree of stenosis [131].

### 5.5. Stroke

The association between Lp(a) and stroke was first reported in the 1980s [132]; since then, a large number of studies have investigated the role of Lp(a) as a risk factor of cerebrovascular disease. In a Danish study that analyzed a large population of about 50,000 individuals [133], high Lp(a) levels correlated with an increased incidence of ischemic stroke, although the specific mechanisms are not fully understood. There is no evidence of different Lp(a) accumulation in carotid artery plaques compared to coronary ones [134], although events triggered by intracerebral vessels or hemorrhagic stroke were associated with lower levels of Lp(a) [126]. The different biology of larger lesions (inflammatory cells’ prevalence, fibrous cap erosion, vessel wall frailty, and superimposed thrombosis) compared to smaller ones and the occurrence of non-atherosclerotic lesions might account for such a discrepancy. Several hypotheses have been advanced to explain the pathophysiology: high levels of Lp(a) could increase the deposition of cholesterol in vessels, while the Apo(a) component, via interfering with fibrinolysis and because of its known proinflammatory properties, could predispose them to plaque. However, the role of Lp(a) in the genesis of other types of stroke is not yet fully clarified: a meta-analysis of Kumar et al. [135], which analyzed the relationship between Lp(a) levels and various subtypes of stroke, shows an association between Lp(a) and stroke secondary to large-vessel atherosclerosis rather than small-vessel and cardioembolic stroke.

There are conflicting results concerning the association between Lp(a) levels and thromboembolic risk in atrial fibrillation (AF) [136]. Igarashi demonstrated that Lp(a) is an independent risk factor for left-atrium thrombosis in patients with chronic AF [137]; more recently, a correlation between elevated Lp(a) levels and thromboembolism in patients with AF and a CHADVASC of less than 2 has been found [138]. Conversely, other works do not find a correlation between Lp(a) levels and cardioembolic stroke [139], suggesting that the different phenotypes of Lp(a) may be involved [140]. Reports are also conflicting with regard to lacunar stroke. Kario et al. demonstrated an association between high levels of Lp(a) and multiple lacunar strokes [141], while other studies [142] have failed to show this correlation.

### 5.6. Heart Failure

Heart failure (HF) is a global public health problem and a major burden, particularly for economies and health systems of countries with aging populations. As mentioned earlier, Lp(a) is an independent risk factor for CAD and calcific aortic valve stenosis, both of which are underlying causes of heart failure (HF) (see Figure 2). In 2016, a study reported an association between Lp(a) and corresponding Lp(a) risk genotypes with HF. Two cohort studies in the combined Copenhagen General population have investigated the correlation between Lp(a) and HF-related outcomes, revealing that the risk of incident HF increases with rising levels of Lp(a) [143]. These findings were subsequently supported by a paper from the ARIC investigators [144]. In the Copenhagen studies, while the increased risk for HF appeared to be mostly driven by CAD and AS, the association between Lp(a) and HF remained significant even after the exclusion of patients with previous MI or AS [145]. This observation implies that Lp(a) might have a part to play in HF via means of alternative pathophysiological mechanisms besides the two mentioned earlier. These may include arterial stiffness related to atherosclerosis or vascular noncompliance, which could result in heightened cardiac afterload.

The association between HF and Lp(a) levels seems to vary according to the ethnic group studied. In a multiethnic cohort, Lp(a) was found to be a significant risk factor for incident HF in Caucasian individuals alone, as in the MESA study [146], and in the Chinese population alone [147]. Recently, a large Icelandic case–control study found an association between Lp(a) genetic variants with risk of HF [148].

Figure 2 displays adjusted hazard ratios for selected outcomes, comparing participants with high Lp(a) levels to those with lower concentrations, in relation to their risk for various health conditions and mortality. The study, conducted as part of the Copenhagen studies, found an association between high Lp(a) levels and an increased risk of CVD as well as mortality. The study calculated hazard ratios by comparing individuals in the upper percentiles of Lp(a) distribution, specifically the 90–95th percentile (yellow) and >95th percentile (red) for aortic stenosis, 91–99th percentile (yellow) and >99th percentile (red) for MI and HF, and >95th percentile for ischemic stroke, cardiovascular mortality, and all-cause mortality, with those who had lower Lp(a) concentrations, which were below the 22nd percentile for calcific AS, below the 34th percentile for MI, HF, and PAD, and below the 50th percentile for ischemic stroke, cardiovascular mortality, and all-cause mortality. The figure presents an adjusted odds ratio for participants in the >66th percentile compared to those in the <33rd percentile for PAD, which was defined as an ankle–brachial index of ≤0.9. It should be noted that the figure was adapted from a publication by Benoit J. Arsenault and Pia R. Kamstrup in the journal *Atherosclerosis* in 2022 [96].

## 6. Pharmacological Treatment

How is elevated Lp(a) treated? The low effectiveness of lifestyle modifications and traditional lipid-lowering therapy, such as statins, niacin, or CETP inhibitors, has aroused greater interest in searching for new drugs that can reduce plasma Lp(a) levels [149]. Statins, which are commonly used to lower cholesterol, do not affect Lp(a) levels and may even increase them slightly. For example, data from JUPITER (Justification for the Use of Statins in Prevention: An Intervention Trial Evaluating Rosuvastatin) showed an increase in Lp(a) plasma levels of 10–20% in patients treated with rosuvastatin [150]. In a recent study of 3,896 patients treated with various statins (including atorvastatin, pravastatin, rosuvastatin, pitavastatin, and simvastatin/ezetimibe), the mean levels of Lp(a) increased by 11% and OxPL-apoB increased by 24% [151]. Another meta-analysis, involving 5256 patients, found that statins significantly increased plasma Lp(a) levels [152]; however, the underlying mechanisms of this increase are not yet well understood, and further studies are required. Furthermore, ezetimibe treatment, both alone and in combination with statins and lomitapide, did not reduce Lp(a) levels, while the effects of fibrates on Lp(a) levels remain uncertain (Table 2) [153,154].

Over the years, niacin, an essential nutrient involved in the synthesis and metabolism of carbohydrates, proteins, and lipids, has been recognized as an atheroprotective agent because of its capacity to lower the plasma levels of cholesterol, triglycerides, VLDL, and LDLc, as well as being able to raise that of high-density lipoproteins [155]. Niacin has also been reported to reduce the plasma levels of Lp(a) by 38–40% [156]. However, the abandonment of this therapeutic strategy was due to the appearance of side effects, particularly hot flashes, abdominal pain, hepatotoxicity, and the failure to achieve primary cardiovascular event reduction endpoints in two large randomized trials, one with prolonged-release niacin (AIM HIGH) [157] and the other with niacin/laropiprant (HPS2-THRIVE) (Table 2) [158].

Drugs that inhibit CETP — a glycoprotein synthetized mostly by the liver that plays a prominent role in the bidirectional transfer of cholesterol esters and triglycerides (TRG) between lipoproteins [159] — are able to increase HDL-C and also decrease serum LDL-C levels [160]. CEPT inhibitors (CEPTis) are also effective in reducing Lp(a) levels [161]: in particular, torcetrapib therapy decreased Lp(a) by 11%, evacetrapib by up to −40%, and evacetrapib combined with statins by 31%, while dalcetrapib had lower effects on lipids than torcetrapib, evacetrapib, or anacetrapib, and decreased Lp(a) by 5% (Table 2) [162].

PCSK9 inhibitors are human monoclonal antibodies, produced by genetic engineering techniques, that have recently been introduced into clinical practice. These drugs bind and inactivate a particular circulating enzyme that plays a central role in modulating the expression of hepatic receptors for low-density lipoproteins (LDL-R): the proprotein convertase subtilisin/kexin type 9 (PCSK9). The blocking of the PCSK9 protein results in a rapid slowdown of LDL receptors’ turnover with an increase in their number, which leads to a marked reduction in plasma LDL concentration by 50–70% [163]. In addition to the LDL cholesterol-lowering effect, PCSK9 inhibitors are also capable of reducing Lp(a) levels by 25–30% [164]. In particular, a recent study evaluated the effects of alirocumab on serum Lp(a) levels via the use of a pool of data from the Odissey phase 3 studies [165]; these data showed a significant reduction in Lp(a) levels regardless of the initial dose in patients treated with alirocumab and the concomitant use of statins. At 24 weeks of treatment, a reduction of 23% to 27% was observed in those patients who started 75 mg of alirocumab, and one of 29% in patients who received 150 mg of alirocumab. The reduction in Lp(a) levels is independent of race, sex, the presence of familial hypercholesterolemia, basal Lp(a) and LDL-C concentrations, and the use of statins. From a meta-analysis obtained from 10 clinical trials on 3,278 patients, a significant reduction in Lp(a) levels of 24.7% was observed in subjects who received 140 mg of evolocumab every 2 weeks, and 21.7% for subjects treated with a monthly dose of 420 mg [166]; however, it should be noted that both the Fourier and Odissey studies were not specifically designed to enroll and treat patients with high Lp(a) levels [167,168]. Furthermore, the use of these drugs is not currently approved for the specific treatment of isolated hyperlipoproteinemia(a). PCSK9 is also the target of inclisiran, a small interfering RNA (siRNA)-based drug that inhibits the hepatic synthesis of this protein. The ORION 10 trial initially measured a median Lp(a) value of 57 nmol/L, with a reduction rate of 25.6% after 510 days of treatment compared to the placebo group [169]. Similarly, in the ORION 11 study the baseline median Lp(a) level was 42 nmol/L, and the relative reduction was 18.6%. In the ORION 9 investigation, although the initial median Lp(a) concentration was 57 nmol/L, the reduction rate was lower, at −17.2% (Table 2) [170].

Mipomersen, a second-generation antisense oligonucleotide (ASO), was approved by the FDA for use in addition to statin therapy to treat homozygous FH [171]. Its mechanism of action involves inhibiting Apo B synthesis without affecting Apo(a). While it can reduce Lp(a) levels by 25–40%, its therapeutic use is limited due to the serious side effects that it can cause, such as site of injection reactions, hepatic steatosis, and hypertransaminasemia (Table 2) [172].

**Table 2 ijerph-20-06721-t002:** The table summarizes different treatment options for elevated Lp(a) levels and their effects on reducing Lp(a) concentrations.

Treatment	Mechanism of Action	Effect on Lp(a) Levels	Notable Findings and Remarks
Statins [150,151,152]	Lowers cholesterol levels	No significant reduction; may increase slightly	Statins do not significantly affect Lp(a) levels and may even lead to a slight increase in some cases. Further research is needed to understand the underlying mechanisms of this increase.
Niacin [156,157,158]	Atheroprotective agent, lowers lipids	Reduction in Lp(a) levels by 38–40%	Niacin has been effective in reducing Lp(a) levels, but its use is limited due to side effects, such as hot flashes, abdominal pain, hepatotoxicity, and a failure to achieve primary cardiovascular event reduction in some trials.
CETP inhibitors [161,162]	Reduces CETP activity, increases HDL	Reduction in Lp(a) levels; varying effects	CETP inhibitors have shown varying effects on Lp(a) levels, with some drugs reducing levels by up to 40%. The effects of fibrates on Lp(a) levels remain uncertain.
PCSK9 inhibitors [164,165,166,167,168]	Blocks PCSK9 enzyme, increases LDL receptor expression	Reduction in Lp(a) levels by 25–30%	PCSK9 inhibitors have been found to be effective in reducing Lp(a) levels by 25–30%, along with lowering LDL-C. Alirocumab has shown promising results in reducing Lp(a) levels independently of factors such as race, sex, FH, and baseline Lp(a) as well as LDL-C concentrations.
Inclisiran [169,170]	Inhibits the hepatic synthesis of PCSK9	Reduction in Lp(a) levels	Inclisiran has shown reductions in Lp(a) levels in clinical trials. The ORION studies demonstrated significant reductions in Lp(a) levels over the course of treatment.
Mipomersen [172]	Inhibits apo-B synthesis without affecting apo(a)	Reduction in Lp(a) levels by 25–40%	Mipomersen can reduce Lp(a) levels but has limited therapeutic use due to serious side effects.
Apheresis [173]	Extracorporeal removal of lipoproteins	Lowering of LDLc and Lp(a) concentrations by 60–70%	Apheresis has been found to be the most efficient and well-tolerated therapy for individuals with Lp(a) hyperlipoproteinemia. It achieves substantial reductions in LDL-C and Lp(a) levels, leading to significant improvements in cardiovascular outcomes, with a 54–90% reduction in cardiovascular events. Apheresis is recommended for patients with progressive coronary disease and Lp(a) levels greater than 60 mg/dL despite maximal lipid-lowering therapy. It has been approved for elevated Lp(a) associated with progressive CVD in Germany. Studies have provided evidence of reduction in cardiovascular risk and long-term efficacy in stabilizing CAD.

Summary of various treatment options for elevated Lp(a) levels and their effects on Lp(a) concentrations. The table provides an overview of different therapeutic approaches, including statins, niacin, CETP inhibitors, PCSK9 inhibitors, inclisiran, mipomersen, and apheresis, along with their respective mechanisms of action and their impact on Lp(a) levels. Notably, statins are found to have limited effectiveness in reducing Lp(a) levels, while niacin shows promise but is hindered by side effects. CETP inhibitors exhibit variable effects, and PCSK9 inhibitors demonstrate significant reductions in Lp(a) levels. Inclisiran, an siRNA-based drug, also lowers Lp(a) concentrations. Mipomersen, an antisense oligonucleotide, reduces Lp(a) levels but faces limitations due to side effects. Apheresis, an extracorporeal method for selective lipoprotein removal, proves to be the most efficient and well-tolerated therapy, achieving substantial reductions in Lp(a) levels, along with LDLc, leading to significant improvements in cardiovascular outcomes.

### 6.1. Lipoprotein Apheresis

Apheresis has been found to be the most efficient and well-tolerated therapy for individuals with Lp(a) hyperlipoproteinemia (Table 2). The data available on lipoprotein apheresis are very impressive, showing a lowering of LDL-C and Lp(a) concentrations by 60–70%, improving the cardiovascular outcomes of these patients with a 54–90% reduction in cardiovascular events [173]. Apheresis is an extracorporeal method of selective removal, from plasma or whole blood, of lipoproteins containing Apo B100, with a reduction of over 50% of atherogenic lipoprotein [174]. The guidelines of the American Society for Apheresis (ASFA) since 2013 have included Lp(a) hyperlipoproteinemia among the indications for lipoprotein apheresis [175]. The HEART UK Lipoprotein apheresis guidelines recommend that apheresis should be considered for those patients with progressive coronary disease and Lp(a) greater than 60 mg/dL whose LDL-C remains 125 mg/dL despite maximal lipid-lowering therapy [176]. In Germany, lipoprotein apheresis has been approved for elevated Lp(a) associated with progressive CVD; since then, the German Lipoprotein Apheresis Registry (GLAR) provides statistical evidence for the assessment of extracorporeal procedures enacted to lower both LDL-C and Lp(a). In a prospective investigation based on GLAR data over a 5-year period, patients with an LDL-C level lower than 100 mg/dL but Lp(a) level higher than 60 mg/dL showed a significant reduction in major coronary (83%) and noncoronary events (63%) [177]. At present, the FDA has authorized the use of lipoprotein apheresis solely for individuals who have documented CVD progression and increased Lp(a) levels exceeding 60 mg/dL [178]. Studies have provided proof of a reduction in cardiovascular risk in patients undergoing regular apheresis; indeed, lipid apheresis achieves near-normal Lp(a) levels and prevents a MACE (major adverse cardiovascular event) [179]. The Prospective Pro(a)LiFe study also supported the notion that the prevention of cardiovascular events is both a rapid and lasting effect of apheresis in patients with progressive CVD associated with Lp(a) hyperlipoproteinemia [180]. Another multicenter study confirmed that long-term treatment with apheresis was at least able to stabilize CAD in most of the individuals with symptomatic elevated Lp(a) [181]. Furthermore, a multicenter retrospective study carried out by the G.I.L.A. (Gruppo Interdisciplinare Aferesi Lipoproteica) with the aim of analyzing the incidence of adverse cardiovascular events before and during lipoprotein apheresis treatment in subjects with an elevated level of Lp(a) (>60 mg/dL), chronic ischemic heart disease, and maximally tolerated lipid-lowering therapy as well as chronic ischemic heart disease, confirmed that lipoprotein apheresis carried long-term efficacy in terms of cardiovascular morbidity [182].

### 6.2. Innovative Strategies

Currently, antisense oligonucleotides (ASOs), small interfering RNAs (siRNAs), and microRNAs—all drugs that aim to reduce Lp(a) by targeting RNA molecules, regulating gene expression, and modulating protein production—are the most widely explored perspectives in this field. Recent trials involving ASOs, capable of inhibiting the expression of Apo(a), were the first randomized trials designed to assess an Lp(a)-lowering therapy [183,184]. Anti-Apo (a) ASOs, injected subcutaneously, bind to plasma proteins and are captured in the liver, where they bind to their target mRNA. Through this mechanism, the assembly of Lp(a) is blocked and the plasma levels of Lp(a) are reduced by more than 80% [185]. Two clinical trials recently revealed promising results according to the direct inhibition of Apo(a) synthesis by ASOs [185,186]: in the first trial, IONIS-APO(a)Rx was administered subcutaneously at dosages of 100 mg, 200 mg, and 300 mg once weekly for 4 weeks at each dose sequentially in patients with high Lp(a) levels in order to analyze the safety and efficacy of the ASO IONIS-APO(a)Rx. Subjects treated with 125–437 nmol/L and ≥438 nmol/L showed, respectively, a 62.8% and 67.7% reduction in Lp(a) concentrations compared with the placebo group [187].

A subsequent randomized, double-blind, placebo-controlled, and dose-ranging trial investigated the decrease in Lp(a) levels at different doses and intervals of IONIS-APO(a)Rx-Lrx [188]: a significant dose-dependent lowering of Lp(a) levels in all of the doses tested was observed. The highest cumulative dose (20 mg weekly) reduced Lp(a) by a mean of 80%. Phase 3 of this study is still ongoing.

Pelacarsen is an ASO that inhibits apolipoprotein, significantly lowers direct Lp(a), and has a neutral to mild lowering effect on LDL-C [189]. In a phase II clinical trial, pelacarsen was administered to patients through subcutaneous injections with different dosages, including 20 mg every 4 weeks, 20 mg every 2 weeks, 20 mg every week, 40 mg every 4 weeks, and 60 mg every 4 weeks; the circulating concentrations of Lp(a) were found to decrease by 35%, 58%, 80%, 56%, and 72%, respectively. Additionally, the side effects observed were mild and rare, with most being limited to reactions at the injection site [190]. Currently, a phase III study is ongoing and is set to end in 2024. This study aims to evaluate the effect of pelacarsen on cardiovascular endpoints in patients with Lp(a) levels ≥ 70 mg/dL who had previously experienced a cardiovascular event within the last 10 years. Participants are being treated with 80 mg of pelacarsen once per month, or a placebo [190].

The last promising frontier in reducing Lp(a) is siRNA. SiRNA functioning is based on post-transcriptional gene silencing: siRNA molecules are usually specific and efficient in the knockdown of disease-related genes [191], therefore reducing the production of a protein of interest. There are two siRNA-based Lp(a)-directed therapies in clinical development targeting Lp(a) mRNA—OLp(a)siran and SLN360—and both have been shown to lower Lp(a) plasma levels by up to 90% [192].

OLp(a)siran reduces Lp(a) levels by directly inhibiting Lp(a) messenger RNA translation in hepatocytes. Its effectiveness in reducing plasma Lp(a) concentration has been demonstrated in several clinical trials. In a phase 1 dose escalation study, participants tolerated a single dose of OLp(a)siran well, and experienced a reduction in Lp(a) concentration ranging from 71% to 97%. These effects persisted for several months after the administration of doses of 9 mg or higher [193]. Promising results were also obtained in phase 2 studies [194,195,196], which confirmed the efficacy of OLp(a)siran in reducing Lp(a) levels. These studies support the use of hepatocyte-targeted siRNA as a viable approach to reducing Lp(a) levels in individuals with an elevated plasma Lp(a) concentration.

Another promising siRNA is SLN360: this drug was tested in vitro for Lp(a) knockdown in primary hepatocytes and it specifically reduced Lp(a) expression in primary human hepatocytes with no relevant off-target effects; a sizeable (up to 95%) and long-lasting (≥9 weeks) reduction in serum Lp(a) was observed [197]. In a phase 1 study that involved 32 participants with elevated Lp(a), SLN360 was well tolerated; the trial confirmed that a dose-dependent lowering of plasma Lp(a) concentrations was observed [198]. These findings warrant further investigations to determine the safety and effectiveness of this siRNA (Table 3).

## 7. Conclusions and Perspectives

Genetic and observational evidence support a causal role of lipoprotein(a) in the development of CVD, PAD, CAVS, and HF. Hyperlipoproteinemia(a) represents a widespread health problem in the global population. The inter-individual variation in plasma concentrations of Lp(a) is large in the general population, and ranges from <1 mg/dL to >1000 mg/dL. Concentrations also differ according to ethnicities, with higher concentrations found in individuals of African descent when compared to populations with European or Asian heritage.

The accurate measurement of Lp(a) levels is crucial and requires the use of validated assays with traceability to ensure consistent cut-offs for high concentrations and proper risk assessment. The advancements in genotyping technologies over the past 15 years have allowed for the identification of genetic variations at the Lp(a) locus that are significantly linked to various vascular diseases. The genetically predicted levels of Lp(a) are consistently associated with CAD, calcific aortic valve stenosis, PAD, and, to a lesser degree, carotid atherosclerosis as well as HF. Therefore, Lp(a) is an essential component of residual cardiovascular risk and should be carefully monitored in patients with a history of cardiovascular disease or those at a high risk of developing it.

Lp(a) is minimally responsive to lifestyle or behavior changes, while other lipoprotein concentrations are affected by these factors, suggesting that Lp(a) levels are mostly genetically determined. The low effectiveness of lifestyle modifications and traditional lipid-lowering therapies, such as statins, niacin, or CETP inhibitors, has aroused greater interest in searching for new drugs that can reduce plasma Lp(a) levels. In addition to the LDL cholesterol-lowering effect, PCSK9 inhibitors are also capable of reducing Lp(a) levels by 25–30%. Another option is inclisiran, a siRNA that performs reductions in LDL and Lp(a) concentrations similar to those of PCSK9. It is also worth considering recently discovered novel therapies, such as ASOs, which reduce Lp(a) plasma levels by 60–80%. The last promising frontier in reducing Lp(a) is siRNA, which has been shown to lower Lp(a) plasma levels by to 90%. Apheresis is considered the most efficacious and well-tolerated treatment option for individuals with elevated levels of low-density lipoprotein cholesterol (LDLc) and Lp(a), as it can lead to a significant decrease of 60–70% in both LDL-C and Lp(a) concentrations. This reduction in lipid levels has been shown to result in a significant improvement in cardiovascular outcomes, with a 54–90% decrease in cardiovascular events observed in these patients.

## Figures and Tables

**Figure 1 ijerph-20-06721-f001:**
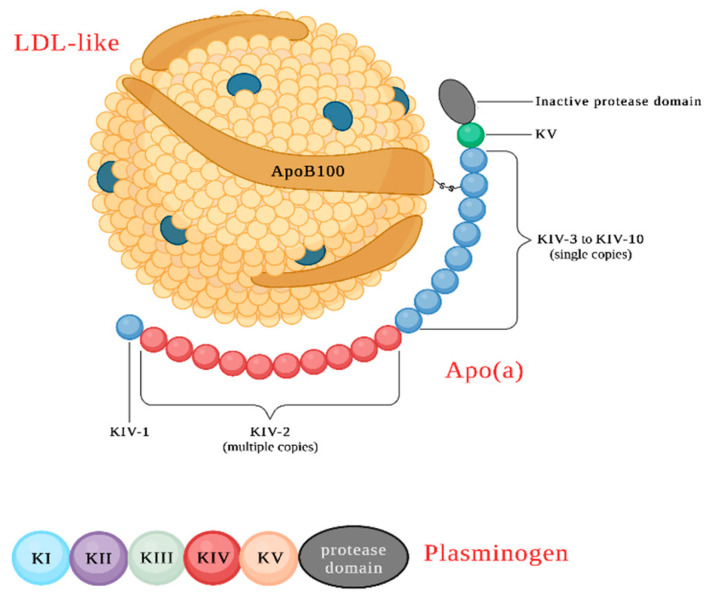
Structure and composition of Lp(a). Lp(a) consists of a low-density lipoprotein to which Apo(a) is added, forming a disulfide bridge between Apo B100 and Apo(a). The protein structure of Apo(a) consists of numerous functional domains. Taken as a whole, it closely resembles the structure of plasminogen, a proenzyme whose conversion into plasmin determines the activation of thrombolytic processes. Plasminogen also exhibits a series of kringle (K) structures, numbered I through V, followed by a serine protease-like catalytic domain that can be cleaved via tissue plasminogen activator and urokinase, thus generating plasmin. In apo(a), some domains closely resemble the analogous domains of plasminogen, including the carboxy-terminal catalytic protein domain which, however, cannot be activated as in plasminogen. The peculiar element of apo(a) is the high number of repetitions of kringle IV; however, these repetitions do not give rise to identical structures, but to subtypes of kringle IV, from type 1 to type 10. All of the subtypes are present in a single copy except for kringle IV type 2, whose repetitions are responsible for the length polymorphism of Apo(a).

**Figure 2 ijerph-20-06721-f002:**
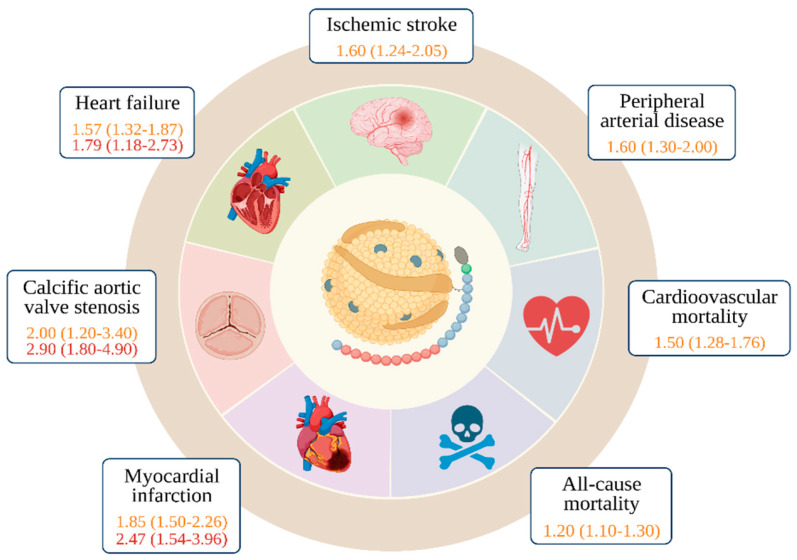
Association between cardiovascular diseases, mortality, and high lipoprotein(a) plasma levels from the Copenhagen studies.

**Table 1 ijerph-20-06721-t001:** Appraised population presenting a lipoprotein(a) plasma concentration >50 mg/dL or >125 nmol/L.

World Area	Prevalence (%)	Prevalence
Asia/China	10%	261 million
Latin America	13%	97 million
Europe	20%	148 million
Australia	20%	8 million
North America	20%	73 million
South Asia	25%	469 million
Africa	30%	376 million
Global	10 to 30%	1.43 billion

**Table 3 ijerph-20-06721-t003:** Clinical trials of RNA targeted therapies to reduce Lp(a).

Clinical Trial Number	NCT03070782	NCT03626662	NCT04270760	NCT04606602
Phase of study	Phase 2	Phase 1	Phase 2	Phase 1
Patient population	286	64	281	32
Tested therapy	Pelacarsen (AKCEA-APO(a)-LRx)	OLp(a)siran	OLp(a)siran	SLN360
Partecipant (drug/placebo)	239/47	48/16 (cohort 1-7)	227/54	24/8
Baseline therapy	80 to 90% of the patients received statin therapy, 50% received ezetimibe, and 20% received a PCSK9 inhibitor	In cohorts 1–5, no participants were on statins; in cohorts 6 and 7, 67% were on statins	88% took statin therapy (including 61% taking high-intensity statin therapy), 52% ezetimibe, and 23% (PCSK9) inhibitor	Concomitant satin use: P: 63%, 30 mg SLN360: 0, 100 mg SLN360: 33%, 300 mg SLN360: 50%, 600 mg SLN360: 60%
Outcomes	The percent change in the lipoprotein(a) level from the baseline to the primary analysis time point at 6 months of exposure (week 25 or week 27)	Safety and tolerability/ change in Lp(a) concentration	Percent change in the lipoprotein(a) concentration from the baseline to week 36 and at week 48	Safety and tolerability/ change in Lp(a) concentration
Dose	20 mg every 4 weeks, 40 mg every 4 weeks, 60 mg every 4 weeks, 20 mg every 2 weeks, or 20 mg every week, or a physiologic saline placebo	Cohort 1: 3 mg sd (n = 6), cohort 2: 9 mg sd (n = 6), cohort 3: 30 mg sd (n = 6), cohort 4: 75 mg sd (n = 6), cohort 5: 225 mg sd (n = 6), cohort 6: 9 mg sd (n = 9), and cohort 7: 75 mg sd (n = 9)	10 mg every 12 weeks, 75 mg every 12 weeks, 225 mg every 12 weeks, or 225 mg every 24 weeks	30 mg sd (n = 6), 100 mg sd(n = 6), 300 mg sd (n = 6), and 600 mg sd (n = 6)
Baseline concentration of Lp(a)	Median levels ranged from 205 to 247 nmol/L	70–199 nM (cohorts 1–5), ≥200 nM (cohorts 6 and 7)	Median (nmol/L): 260.3 (interquartile range of 198.1 to 352.4)	Median (nmol/L): P 238, 30 mg 171, 100 mg 217, 300 mg 285, 600 mg 231
Percent change from baseline Lp(a)	Decreases of 35% at a dose of 20 mg every 4 weeks, 56% at 40 mg every 4 weeks, 58% at 20 mg every 2 weeks, 72% at 60 mg every 4 weeks, and 80% at 20 mg every week, as compared with 6% for a pooled placebo group	From −71% to −97% in cohorts 1–5 and from −76% to −91% in cohorts 6 and 7	At 36 weeks: + 3.6% in the placebo group, −70.5% with the 10 mg dose every 12 weeks, −97.4% with the 75 mg dose every 12 weeks, −101.1% with the 225 mg dose every 12 weeks, and −100.5% with the 225 mg dose every 24 weeks. At 48 weeks: −68.5% with the 10 mg dose every 12 weeks, −96.1% with the 75 mg dose every 12 weeks, −100.9% with the 225 mg dose every 12 weeks, and −85.9% with the 225 mg dose every 24 weeks	P: −10%, 30 mg: −46%, 100 mg: −86%, 300 mg: −96%, 600 mg: −98%

P, placebo; Sd, single dose.

## Data Availability

Not applicable.

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
