# Peer review of "Lipoprotein(a) as a Risk Factor for Cardiovascular Diseases: Pathophysiology and Treatment Perspectives"

_ijerph, 2023, doi:10.3390/ijerph20186721_

Round 1
Reviewer 1 Report
Vinci's manuscript entitled: “Lipoprotein(a) as a Risk Factor for Cardiovascular Disease: Pathophysiology and Treatment Perspectives is interesting and deserves to be published”
I suggest that the following proposals be reviewed:
Figure 1 is interesting, but it should be better described
The subtitle: 1.2. "The Role of Genetic and "Non-genetic" Factors" should be added and reworded. Please specify the role of what. In plasma concentration?
Line 112, specify which polymorphism(s)
Line 129, after Lp(a) add concentrations
Lines 144-147, add references
Line 152, please add a more specific reference to tobacco use rather than a revision
Line 158, please better explain this statement, " Indeed, "Lp(a) could enhance wound healing"
Line 166, Please specify the role of inflammatory cells: Activation?
Line 296, "if this evidence were confirmed" should be rephrased to " if this evidence will be confirmed
Line 302, add a specific reference after autotaxin (ATX)
Line 311, use the abbreviation for apolipoprotein (a),
Line 314, please reword this sentence: Although elevated Lp(a) is recognized as a significant predictor of clinical aortic stenosis in patients with heterozygous familial hypercholesterolemia (FH), its impact on aortic stenosis in the general population has been established.
Line 321, add a reference after the ones
Line 363, why are you only reporting the incidence for the US? Please expand to other countries
Line 382, please rephrase this sentence: in contrast with the findings above, Ooi and colleagues showed that plasma Lp(a) concentration, but not apo(a) isoform size, estimated by KIV-2 copy number, while independently associated with angiographic extent and severity of coronary artery disease (CAD) wasn't significantly associated with carotid artery plaque.
Line 398, is there preferential accumulation in the carotid arteries? Explain better (see van Dijk RA, Kolodgie F, Ravandi A, et al. Differential expression of oxidation-specific epitopes and apolipoprotein(a) in progressing and ruptured human coronary and carotid atherosclerotic lesions. J Lipid Res 2012;53: 2773-90.)
Paragraph 5 "Pharmacological Treatment," is of particular interest. Please summarize the studies listed in a table
Line 521, delete "According to research."
Moderate editing of English language
Author Response
Dear Reviewer thank you for your suggestions that will improve the quality of the manuscript
please find enclosed the revised version of the manuscript

Reviewer 2 Report
1. The manuscript lacks Introduction and explanatory part as to why this review was undertaken
2. The need for this review is unclear
3. What were the objectives of doing this review
4. Some parts of the review I think are already established without any novelty like Aortic Valve Stenosis pathophysiology is well understood , how does this review adds to the current knowledge
5. Search Strategy not defined
6. Conclusion with the table looks obscure
Minor spell check
Author Response

(The authors gave the same response as above.)
